# Assessing the Impact of Microwaves and Other Disruptive Pretreatments on *Lactiplantibacillus plantarum* Growth and the Antioxidant Properties of Broccoli Stalks

**DOI:** 10.3390/foods14101809

**Published:** 2025-05-20

**Authors:** Simone Baldassa, Cristina Barrera, Marta Muñoz-Ibáñez, Lucía Seguí

**Affiliations:** 1Dipartamento di Biomedicina Comparata e Alimentazione, Università degli Studi di Padova, Viale dell’Università, 16, 35020 Legnaro, Padova, Italy; baldassa.simone@libero.it; 2Institute of Food Engineering—FoodUPV, Universitat Politècnica de València, Camino de Vera, s/n, 46022 Valencia, Spain; mmuoiba@upv.edu.es (M.M.-I.); lusegil@upvnet.upv.es (L.S.)

**Keywords:** broccoli residues, plant-based fermentation, LAB fermentation, thermophysical pretreatments, probiotic

## Abstract

Food waste is a global challenge, with broccoli stalks (~58% of the head’s mass) often discarded despite being rich in fiber, fatty acids, organic acids, sugars, phenolic compounds, and glucosinolates. Pretreatments like microwaving are gaining interest for enhancing nutrient availability and microbial activity by modifying biomass structure and improving fermentation. This study explores microwave pretreatment (2, 4, 6, 9 W/g for 4–7 min) to enhance 24 h fermentation of pasteurized ground broccoli stalks using *Lactiplantibacillus plantarum*. Analyses included reducing sugars, total phenolics, microbial growth, Cryo-FESEM, and FTIR. Optimal microbial proliferation and preservation of key compounds were achieved at 4 W/g for 5 min. This condition was then compared to pasteurization, freezing/thawing, and autoclaving over a 96 h fermentation. pH, microbial population, and antioxidant properties were measured at 24 h intervals. Pasteurization, with or without microwaving, resulted in faster acidification and microbial growth. Microwaved samples retained the highest phenolic content, while autoclaved ones exhibited the highest flavonoid levels and antioxidant activity. Fermentation did not enhance antioxidant properties; phenolics and DPPH activity decreased after 24 h, while flavonoids and ABTS remained stable. Overall, pretreatments significantly influenced fermentation outcomes of broccoli stalks; microwaving post-pasteurization favored phenolic preservation and microbial proliferation, whereas autoclaving enhanced flavonoids and antioxidant potential.

## 1. Introduction

The reduction in food waste is one of the biggest challenges that agricultural and food systems have been facing in recent years. In 2022, only in Europe, 59 million tons of fresh mass were lost, with more than half (53%) being wasted at the household level [1]. The Food and Agriculture Organization of the United Nations (FAO) reported that 14% of the world’s food production was wasted during the production steps (post-harvest and supply chain, excluding retailing) [2]. According to FAO estimates, the food that is lost and wasted could feed 1.26 billion hungry people every year. The concept of inedibility plays an important role in household food waste, whereas in industrial production of ready-to-cook and ready-to-eat foods, edibility is well defined and regulated. When looking at broccoli utilization, especially frozen and IV-range ready-to-cook products, only florets are used to produce the final product, whereas the stalk, leaves and root parts are considered inedible, although they represent the majority of the edible part of the plant. According to the values reported by Li et al. [3], the average composition of broccoli plants (*Brassica oleracea*, var. *italica*) is stalks 21%, leaves 47%, roots 17%, and florets 15% (w/w of fresh mass). Composition varies along the different parts of the plants and, while florets are richer in amino acids, stalks are richer in fatty acids, organic acids (mostly lactic, acid, malic, and maleic acids), and sugars [4]. Broccoli contains several antioxidant substances, with their content also varying among the three main parts of the plant (stem, floret, and leaves). Valorizing the normally excluded tissues as a source of antioxidants can represent an effective solution to reduce waste during the processing step of this vegetable and, with that, reduce the loss of nutrients. Several antioxidant molecules have been identified in broccoli [3,5]. In brassica vegetables, glucosinolates and isothiocyanates stand out, of which the sulfhydryl group can act as an antioxidant while stimulating conjugating enzymes (phase II enzymes of compounds metabolization) and phenols. Despite this potential, broccoli stalks remain significantly underutilized in food systems and are among the most common vegetable by-products. Developing scalable strategies to valorize these tissues, such as through fermentation, is crucial in the transition towards a more circular and sustainable food economy.

Several studies already explored the possibility of utilizing physical, chemical, and thermal treatments to enhance the concentration of various molecules by degrading more complex molecular structures, such as cellular membranes and complex polymers [6,7,8,9]. All major cooking methods, like freezing, boiling, microwaving, and blanching, produce some kind of structural damage to vegetable tissues, like cell disruption and parenchyma rupture, causing decrease in turgor, loss of water, and loss of texture [10]. Tissue damage can be linked to the release of some antioxidant molecules [11,12]. Antioxidant extraction can be promoted or intensified by the use of physical treatments that may induce cell breakage, such as ultrasounds and microwave-assisted extraction [13]. The stability of different antioxidant molecules has a significant influence on how these treatments affect the antioxidant properties of the matrix. In broccoli, for instance, while some authors reported that cooking methods such as boiling and microwaving led to a reduction in antioxidant activity [14], others observed that microwaving between 3 and 5 min can effectively increase the concentration of phenols and the antioxidant activity [15]. The conflicting results regarding the effect of pretreatments on antioxidant properties will be further explored in this study.

Microwaves have been suggested among the alternative methods for biomass pretreatment. Different studies have investigated microwave positive effects on biomass to improve further bioconversion [9,16]. Microwaves interact with polar molecules and ions present in the waste material, resulting in both thermal and non-thermal effects that modify the properties of the biological material and have a positive impact on reaction rates [17]. Thermal effects are responsible for temperature and subsequent pressure increase, which implies disruption of lignin and cell walls, an effect that has been claimed to facilitate further enzyme action. Therefore, microwave irradiation may generate explosions within a material’s particles, resembling hydrothermal treatments, thereby easing the disruption of the recalcitrant structures [9,18]. On the other hand, non-thermal effects are the result of dielectric polarization of covalent bonds due to microwave irradiation, which may also disrupt biomass components such as lignin, cellulose, and hemicellulose [18,19]. Structural analysis and microscopic observations of microwave pre-treated biomass have shown structural changes, which facilitate further enzyme action such as reduced particle size, reduced density, and increased tissue porosity and surface area, or evidence lignin degradation [9,20,21]. However, there is still limited data on how microwave intensity and duration specifically influence plant structure, fermentation outcomes, and the transformation of bioactive compounds. Other physical treatments that promote cell disruption, such as grinding or freeze–thawing, can also be proposed to enhance bioconversion by increasing simple, fermentable sugars, such as glucose and fructose, which can favor enzyme efficiency and microbial activity. On the other hand, thermal treatments, such as pasteurization, and hydrothermal ones, such as autoclaving, which imply high temperatures and high pressure, modify the plant structure so that subsequent fermentation processes can be facilitated. Given the growing interest in sustainable bioprocessing and the development of functional ingredients, a more detailed understanding of how commonly used pretreatments influence fermentation outcomes is needed.

Brassica plants, cabbages in particular, are among the most fermented vegetables in the world. The benefits of this transformation are largely known and include a wide range of positive effects, such as improved organoleptic characteristics, improved preservability of the product, recreational purposes (i.e., alcoholic beverages), hydrolysis of complex molecules and health-related characteristics, like the action of probiotic bacteria [22], and the production of bacteriocins [23]. *Lactiplantibacillus plantarum* (together with other strains like *Levilactoplantibacillus brevis* and *Limosilactoplantibacillus fermentum*) have even shown the ability to enhance the bioactivity of some molecules such as glucosinolates and polyphenols, which require activation through digestion by digestive enzymes that may not always fully hydrolyze them [24]. *Lactiplantibacillus plantarum* can also produce bacteriocins, generally referred to as plantaricins, as well as various antifungal compounds, including polypeptides, proteinaceous substances, and certain organic acids [25]. It has been classified as probiotic microorganism since 1999 [26].

The aim of the present study was to assess the efficiency of a series of physical and thermal pretreatments, including freeze–thawing (FR/TH), microwaving (MW), pasteurization (PAST), and autoclaving (AUTO), in enhancing the fermentation of broccoli waste by lactic acid bacteria and improving the antioxidant characteristics of the fermented residue. In the first part of the study, different microwave pre-treatments are applied to pasteurized ground broccoli stalks and are selected based on microbial growth, changes in reducing sugars, and total phenolic content after 24 h of fermentation with *Lactiplantibacillus plantarum*. In the second part, the optimal microwave power and time combination are applied, and its impact on microbial counts and antioxidants properties along 96 h of fermentation with *L. plantarum* are assessed and compared to other thermophysical pretreatments.

## 2. Materials and Methods

### 2.1. Raw Material and Microbial Strain

Fresh broccoli heads (*Brassica oleracea* var. *Italica*) were purchased from a local supermarket and stored at 4 °C for a maximum of 24 h before use. The first step in sample preparation involved separating the stalks from the florets with a knife, followed by cutting the stalks into 1 cm × 1 cm pieces. The cubed broccoli stalks were then washed with a 0.37% (*w*/*v*) sodium hypochlorite commercial solution at a 1/10 (*v*/*w*) ratio, followed by rinsing with tap water. Once sanitized, the broccoli stalks were ground for 4 s at maximum speed using a TM31 Thermomix^®^ food processor (Vorwerk, Madrid, Spain) in 200 g batches. The resulting mash was transferred into sterilized 250 mL twist-off glass jars and pasteurized in a JP Selecta^TM^ Precisdig thermal bath at 80 °C (ThermoFisher Scientific, Waltham, MA, USA) until an internal temperature of 72 °C was reached and maintained for 1 min.

*Lactiplantibacillus plantarum* CECT 749 (Colección Española de Cultivos Tipo, Paterna, Spain) was selected as the microbial strain for fermentation. The choice was based on its probiotic potential [27], its Qualified Presumption of Safety (QPS) status [28], and its adaptability to various ecological environments [29]. The selection was also supported by our previous findings, in which *L. plantarum* CECT 749 outperformed other probiotic strains, such as *L. reuteri* CECT 925 and *L. salivarius* spp. *salivarius* CECT 4063, achieving the highest microbial counts in both ground and chopped broccoli stems after 24 h of fermentation [30]. When necessary, the frozen suspended culture was regenerated in MRS broth (SharlauChemie^®^, Barcelona, Spain) by incubating for 24 h at 37 °C in a JP Selecta Incudigit (Barcelona, Spain) incubator. Subsequently, 0.1 mL of the culture was plated undiluted onto MRS agar (SharlauChemie^®^, Barcelona, Spain) plates and incubated at 37 °C for at least 24 h, or until a visible microbial layer had formed. The plates were then stored at 4 °C until use. To prepare the final inoculum, an adequate amount of bacterial culture was collected, diluted in 9 mL of MRS broth, and incubated at 37 °C for 24 h until a population of (6 ± 3) × 10^9^ CFU/mL was achieved.

### 2.2. Thermophysical Pretreatments

Several microwave conditions for enhancing fermentation efficiency were tested using a domestic microwave unit (Samsung GW72N): 100 W/50 g (2 W/g), 300 W/75 g (4 W/g), 300 W/50 g (6 W/g), and 450 W/50 g (9 W/g) for 4, 5, 6, and 7 min, and 600 W/50 g (12 W/g) for 2, 3, and 4 min. Power-to-mass ratios and exposure durations were selected based on preliminary trials and relevant studies [9] to span a broad range of thermal inputs without inducing excessive browning or degradation. Sample weights were measured before and after microwaving and subsequent cooling, so that the water lost during the treatment was restored post-treatment to maintain consistency. The added water was previously sterilized to avoid contamination with unintended microbial strains and was reintegrated into the plant tissue using a sterile spatula to ensure even distribution before sampling and further inoculation. Selection criteria included total reducing sugar content, total phenolic content (measured before and after 24 h fermentation), and the ability of *Lactobacillus plantarum* CECT 749 to grow on ground broccoli stalks. Additionally, microscopic observations and FTIR spectra were evaluated.

In the second part of this study, additional thermal and physical pretreatments were investigated and compared to the most efficient microwave (MW) pretreatment. These included freezing at −20 °C for 24 h followed by thawing and pasteurization (FR/TH), autoclaving (AUTO) at 121 °C and 1.1 ± 0.1 bar for 5 min using a Systec GmbH VB-40 autoclave (Göttingen, Germany), and pasteurization (PAST) at 72 °C for 1 min as a control. Analytical determinations included pH, total phenolic and flavonoid content, DPPH and ABTS free radical scavenging ability, and microbial counts over a 96 h fermentation with *Lactiplantibacillus plantarum* CECT 749.

Flow diagrams illustrating the sequence of unit operations and the analytical determinations performed during both the preliminary study on microwave pretreatment and the comparative study of thermophysical pretreatments are shown in Figure 1.

### 2.3. Inoculation and Fermentation

Fermentation was conducted in 250 mL glass jars sealed with twist-off closures. Each jar contained 200 g of ground broccoli stalks, to which 1 mL/100 g of the previously described inoculum was added and mixed thoroughly. The jars were incubated at 37 °C in a JP Selecta Incudigit (Barcelona, Spain) incubator. For the microwave pretreatment (MW) testing, fermentation was carried out over a period of 24 h. In comparative tests involving the selected combination of microwave power and time (MW), autoclaving (AUTO), freezing and thawing (FR/TH), and pasteurization (PAST), fermentation was monitored by plate counting every 24 h for up to 96 h.

### 2.4. Analytical Determinations

#### 2.4.1. Moisture Content and pH

Water content (x_w_) was determined gravimetrically by measuring the weight loss of a known amount of sample dried using a two-step procedure, adapted from the AOAC official method 934.06 [31]—first, in a stove (JP Selecta Conterm, Barcelona, Spain) at 60 °C and atmospheric pressure for 24 h, and second, in a vacuum oven (JP Selecta Vaciotem, Barcelona, Spain) at 0.2 bar and 60 °C until a constant weight was achieved.

The pH was measured at 25 °C in a beaker containing 2.5 g of sample suspended in 50 mL of distilled water using a SevenDirect SD20 pH meter (Mettler Toledo International Inc., Columbus, OH, USA).

#### 2.4.2. Reducing Sugar Content

Reducing sugar content was measured using the dinitrosalicylic acid (DNS) method with some modifications [32]. Methanolic extracts were prepared by mixing 0.3 ± 0.001 g of sample with 2 mL of an 80% (*v*/*v*) ethanol in water solution. The mixture was vortexed and allowed to rest for 30 min, followed by centrifugation at 10,000 rpm for 10 min using a MiniSpin^®^ mini centrifuge (Eppendorf AG, Hamburg, Germany). The resulting precipitate was then re-extracted with an additional 2 mL of solvent. For the DNS assay, 0.5 mL of the extract was mixed with 1 mL of DNS reagent (1 g of 3,5-dinitrosalicylic acid in 100 mL of a solution containing 30 g of potassium sodium tartrate and 16 g of NaOH). The tubes were then placed in a thermal bath at 100 °C for 5 min. Then, 6 mL of cold distilled water was added, and the absorbance was measured in a Helios zeta UV/VIS spectrophotometer (Thermo Fisher Scientific Inc., Waltham, MA, USA) at 546 nm. A calibration curve from 0 to 10 mg/L was previously prepared using D-glucose (HPLC grade) in distilled water, allowing the results to be expressed as mg of D-glucose per g of dry weight (mg GE/g_dw_).

#### 2.4.3. Antioxidant Properties

All antioxidant-related analysis required an extraction step, which involved mixing 4 g of the sample with 10 mL of an 80% (*v*/*v*) methanol-in-water solution. The mixture was shaken in the dark for 1 h on a WY-100 laboratory orbital shaker (Comecta GmbH, Kiel, Germany), then centrifuged at 10,000 rpm and 4 °C for 5 min using a 5804 R multipurpose centrifuge (Eppendorf AG, Hamburg, Germany). Only the liquid phase was used for analysis.

Total phenolic content (TPC) was analyzed through the Folin–Ciocalteu method, as previously described by Singleton et al. [33], with minor modifications. A 0.125 mL aliquot of the extract was mixed with 0.5 mL of bidistilled water and 0.125 mL of Folin–Ciocalteu reagent (Scharlab S.L., Barcelona, Spain) and allowed to react in the dark for 6 min. Subsequently, 1.25 mL of a 7% (*w*/*v*) sodium carbonate solution and 1 mL of bidistilled water were added. After incubating in darkness for 90 min, absorbance was recorded at 760 nm using a Helios Zeta UV/Vis spectrophotometer (Thermo Fisher Scientific Inc., Waltham, MA, USA). The absorbance data were converted to mg of gallic acid per g of dry weight (mg GAE/g_dw_) using a standard curve prepared with gallic acid concentrations ranging from 0 to 600 mg/L.

Total flavonoid content (TFC) was determined using a modified aluminum chloride assay based on Luximon-Rama et al. [34]. A 1.5 mL extract aliquot was mixed with 1.5 mL of a 2% (*w*/*v*) aluminum chloride solution in methanol and incubated in the dark for 10 min. Absorbance was then measured at 368 nm using a Helios Zeta UV/Vis spectrophotometer (Thermo Fisher Scientific Inc., Waltham, MA, USA). The absorbance data were converted to mg of quercetin equivalents per g of dry weight (mg QE/g_dw_) using a standard curve prepared with quercetin concentrations ranging from 0 to 200 mg/L.

Antioxidant activity was assessed using the DPPH and ABTS radical scavenging methods. The ABTS method was conducted following the procedure described by Re et al. [35]. First, the pre-formed radical monocation of 2,2′-azinobis-(3-ethylbenzothiazoline-6-sulfonic acid) (ABTS^•+^) was generated by preparing an aqueous solution containing 0.06% (*w*/*v*) potassium persulfate and 0.38% (*w*/*v*) ABTS, which was kept in darkness at 4 °C for at least 16 h, or until needed. Subsequently, the resulting solution was diluted with a potassium buffer containing 2.58% (*w*/*v*) sodium phosphate monobasic anhydrous (NaH_2_PO_4_) and 11.54% (*w*/*v*) disodium hydrogen phosphate anhydrous (Na_2_HPO_4_) until an absorbance of 0.70 ± 0.02 was achieved at 734 nm. A volume of 2.9 mL of this diluted solution was then mixed with 0.1 mL of the extract aliquot, and the absorbance was measured at 734 nm using a Helios Zeta UV/Vis spectrophotometer (Thermo Fisher Scientific Inc., Waltham, MA, USA) after a 7 min reaction time.

Following the DPPH method described in Brand-Williams et al. [36], 0.1 mL of the extract was combined with 2.9 mL of a 0.1 mM DPPH (2,2-diphenyl-1-picryl hydrazyl) solution in methanol. After a 60 min reaction in the dark, the absorbance was measured at 575 nm using a Helios Zeta UV/Vis spectrophotometer (Thermo Fisher Scientific Inc., Waltham, MA, USA).

The absorbance measurements obtained from the ABTS and DPPH assays were expressed as milligrams of Trolox equivalents per gram of dry weight (mg TE/g_dw_). This conversion was based on a standard curve constructed from serial dilutions of 6-Hydroxy-2,5,7,8-tetramethylchroman-2-carboxylic acid within the concentration range of 0 to 0.2 mg/L.

#### 2.4.4. Cryo-Field Emission Scanning Electron Microscopy (Cryo-FESEM)

Tissue damage induced by microwave pretreatment was evaluated using the Cryo-FESEM technique. The analyses were conducted at the Servicio de Microscopía of the Universitat Politècnica de València, utilizing a Zeiss Ultra 55 microscope (Carl Zeiss Microscopy GmbH, Oberkochen, Germany) in conjunction with a Quorum PP3010 sublimation chamber. Samples were affixed to the analytical plate using a cement made by mixing colloidal graphite in water (G303 Colloidal Graphite AQUADAC, Oxford Instruments plc, Abingdon, Oxfordshire, England) and a tissue fixative (Tissue-Tek AutoTEC^®^ a120, Sakura Finetek USA, Inc., Torrance, CA, USA). Conductive strings of cement were created to enhance the distribution of electron flux.

Vegetal tissue was freeze-dried under vacuum conditions using liquid nitrogen (−196 °C) prior to fracturing and drying in the sublimation chamber at −90 °C and 10^−7^ mbar for 15 min. Sputtering was performed at 5 mA using platinum for 20 s. Finally, the samples were observed on a monitor equipped with ZEISS SmartSEM software (version 5.06 with Service Pack 4 ) at a temperature of −150 °C, with an acceleration voltage between 10 and 20 kV.

#### 2.4.5. Fourier-Transform Infrared Spectroscopy (FTIR)

FTIR spectra of broccoli samples treated with different power and time combinations were recorded in attenuated total reflectance (ATR) mode at room temperature using a Cary 630 FTIR spectrophotometer (Agilent Technologies, Santa Clara, California, USA). The spectra were obtained with a resolution of 4 cm^−1^ over a wavelength range of 650–4000 cm^−1^, with a minimum of 32 scans per sample. For normalization, the maximum (I_max_) and minimum (I_min_) spectral intensities were identified, and the normalized values (I_min-max_) were calculated using Equation (1), as suggested by Agustika et al. [37].(1)$$I_{min-max}=\frac{I_{i}-I_{min}}{I_{max}-I_{min}}$$

#### 2.4.6. Microbial Counts

Colony counts of *Lactiplantibacillus plantarum* were determined through serial dilution in peptone water up to 10^−8^, followed by plating 0.1 mL onto MRS agar plates. The plates were incubated for 24 to 48 h at 37 °C in a JP Selecta Incudigit (Barcelona, Spain) incubator. The first dilution of solid samples was prepared by homogenizing 3 g of the sample with 27 mL of sterile peptone water using a stomacher (Interscience model BagMixer^®^ 400, Saint Nom la Bretêche, France). After incubation, the colonies on the plates containing between 30 and 300 colony forming units (CFUs) were counted.

### 2.5. Statistical Analysis

All data were based on three independent experimental units with two technical replicates each, and results were expressed as mean ± standard deviation. Statistical analysis of the results obtained was carried out using Statgraphics Centurion software (Centurion XVII.I version, StatPoint Technologies, Inc., Warrenton, VA, USA). One-way and multifactorial analysis of variance (ANOVA) were performed after verifying data normality, with a 95% confidence level (*p*-value < 0.05). Tukey’s Honest Significant Difference (HSD) test was applied to identify significant differences among groups.

## 3. Results and Discussion

### 3.1. Preliminary Study on Microwave Pretreatment

#### 3.1.1. Impact on RSC, TPC, and Microbial Growth

The impact of pasteurization and subsequent pretreatment with different combinations of microwave power and time on the reducing sugar content (RSC), the total phenolic content (TPC), and the growth ability of *Lactiplantibacillus plantarum* CECT 749 in ground broccoli stalks is presented in Table 1. The RSC value for the raw material was slightly higher than that reported by Rosa et al. [38] for broccoli inflorescences, likely because stems act as storage and transport structures for nutrients, including sugars, supporting plant growth. Similarly, the TPC value obtained for unprocessed broccoli stalks in this study was slightly higher than that reported by Núñez-Gómez et al. [39], which could be attributed to varietal or seasonal differences.

Pasteurization resulted in a 35.4% increase in RSC and a 31.5% rise in TPC compared to raw broccoli stalks. According to Zhang et al. [40], the observed increase in TPC may be attributed to several factors that counteract thermal degradation. These include (a) enhanced extraction efficiency due to cell wall disruption and matrix softening; (b) the release of polyphenols previously bound to dietary fibers; and (c) the effective inactivation of polyphenol oxidase, the enzyme primarily responsible for polyphenol oxidation, thereby minimizing TPC loss through oxidative degradation. The increase in RSC may result from the breakdown of cell wall polysaccharides (e.g., cellulose, hemicellulose, and pectin) into simple sugars due to mechanical disruption or enhanced hydrolysis of glycosylic bonds within these polysaccharides during pasteurization. Additionally, mild heating could temporarily activate endogenous enzymes, such as pectinases, leading to a transient increase in reducing sugar content before enzyme denaturation occurs. This observation is supported by Sew et al. [41], who reported an increase in the residual pectin methyl esterase activity of pineapple juice as the temperature increased from 50 to 60 °C.

Microwave treatment after pasteurization resulted in a general decrease in both RSC and TPC. None of the tested combinations of microwave power and exposure time significantly enhanced the RSC achieved by ground broccoli stalks after pasteurization. On the contrary, treatments at 2 W/g for 7 min, 4 W/g for 4 to 6 min, 6 W/g for 4 to 7 min, and particularly 9 W/g for 6 min led to a statistically significant reduction in RSC compared to the pasteurized sample. While microwave heating was anticipated to disrupt plant cell walls and promote the breakdown of complex carbohydrates, this effect appeared negligible when applied after the pasteurization step. Instead, sugar degradation due to non-enzymatic browning and other heat-induced degradative processes likely dominated [42,43]. Low power and short exposure times (2 W/g for 4 to 6 min) may not have generated any significant effect on the pasteurized samples. Conversely, high power and short exposure times (9 W/g for 4 to 5 min) may have facilitated the degradation of complex carbohydrates but could have also promoted simple sugar degradation, resulting in no significant differences between microwave-treated and pasteurized samples. Hydrothermal degradation of sugars is an extensively documented phenomenon and can occur during microwave treatment, as evidenced in studies on fruit residues [9]. This effect becomes more relevant when increasing power or treatment duration, since the simple sugars released due to local explosions or other phenomena can be subsequently degraded, leading to the generation of compounds which are inhibitory to microorganisms such as furfural derivatives. In the present research, sugars released during pasteurization could have experienced hydrothermal degradation, not being compensated by the new release of simple sugars from complex polysaccharides due to microwave effects when powers such as 4 or 6 W/g were applied.

None of the tested power and time combinations significantly increased TPC compared to pasteurized samples. However, a statistically significant increase in TPC to 3.40 ± 0.10 mg GAE/g_dw_ was observed after microwave treatment at 12 W/g for 4 min, suggesting that these intense conditions may have facilitated the release of bound polyphenols or their transformation into more bioactive forms, effectively counteracting thermal degradation. Additionally, the browning observed when these conditions were applied suggests the formation of Maillard reaction-derived compounds which exhibit antioxidant activity and react with the Folin–Ciocalteu reagent. This phenomenon has been previously observed in broccoli stems and outer leaves of cabbage when air-dried at 70 °C [6]. Calcination phenomena are common in microwave-treated samples when increasing power and/or treatment duration and indicate that temperatures above the boiling point of water are being reached. The appearance of calcination phenomena usually defines the power and time exposure limits for microwave-treated samples [9]. Therefore, this undesirable burning prevented the authors from testing longer treatments at this microwave power. Likewise, microwave treatments causing calcinations or significant browning were not considered for further fermentation (NF in Table 1).

Samples pasteurized and microwaved at 4 W/g presented a significant decrease in TPC compared to pasteurized samples, except for the 5 min treatment. This fact would indicate that under these conditions, microwaves would not promote the release of new phenols but that, instead, it would have a negative impact on the phenols already present or released during pasteurization. Additionally, since the Folin–Ciocalteu reagent can react with reducing sugars [33], these values could also be affected by sugar degradation during microwave heating. Nevertheless, at higher powers and longer exposure times, the potential degradation of phenolic substances and sugars is balanced and overcome by the release of new ones or the conversion to more active forms due to heating [44].

Regarding the microbial population reached after 24 h of fermentation, all treatments demonstrated a significant increase in bacterial counts, rising from an initial load of approximately 7 log CFU/g (inoculated media) to about 9–10 log CFU/g. These results confirm that broccoli stalks serve as an excellent substrate for the growth of *Lactiplantibacillus plantarum* [45,46,47]. The samples treated at 9 W/g exhibited the highest microorganism proliferation, with statistically significant differences as compared to the rest of conditions applied. These samples did not present the highest simple sugar content, for which the increased growth would not be due to fermentable sugar concentration in the substrate but rather to an increased accessibility of microorganisms to the plant matrix constituents. As discussed in the introduction section, this result would be in line with the literature in the sense that microwave pretreatment enhances further bioconversion by promoting structural changes, which make the substrate more susceptible to enzymatic degradation and microbial action [9,48]. As deduced from the present results, at low microwave powers, microbial counts at 24 h of fermentation generally increased alongside the reducing sugar content; in contrast, when increasing the microwave power, the impact on microbial growth would be more determined by the increased cell–matrix accessibility. At intermediate powers, such as 4 W/g, it is observed that the treatments which exhibited lower TPC resulted in higher microbial growth than others with similar RSC but higher TPC. This could be attributed to increased substrate accessibility due to microwave-induced structural degradation. Additionally, it can be hypothesized that a reduced content in phenols, given their known antimicrobial effects [49,50], may have favored microbial growth compared to samples with higher TPC.

Fermentation of plant-based materials with probiotic strains allows us to obtain a probiotic product, but it is also proposed to improve the polyphenol content of the fermented product as a result of microbial metabolism. Polyphenols have well-documented benefits, particularly in preventing diseases associated with oxidative stress [51]. Several studies have documented an increase in TPC and corresponding antioxidant activity following the fermentation of vegetable substrates with lactic acid bacteria. Ye et al. [47] reported a significant increase in total phenolic acid levels (from 289 μg/g to up to 3105 μg/g) after LAB fermentation of autoclaved broccoli puree, primarily due to the generation of phloretic acid by various *L. plantarum* strains. Similarly, Zdziobek et al. [46] reported that the TPC of an 8:11 (*w*/*v*) grated broccoli and water mixture increased after 8 days of fermentation with *L. plantarum* ATCC 8014, rising from 4.17 ± 0.04 mg GAE/mL to 30 ± 5 mg GAE/mL at 30 °C and from 3.88 ± 0.14 mg GAE/mL to 22 ± 2 mg GAE/mL at 35 °C. On the other hand, Salas-Millán et al. [52] found that spontaneous fermentation of broccoli stalk slices in brine at a 2:3 ratio (*w*/*v*) initially increased TPC from 43 ± 2 mg/100 g to 50 ± 3 mg/100 g after 3 days but led to a decrease to 21 ± 5 mg/100 g after 6 days due to the diffusion of phenolic compounds into the brine, as well as enzymatic degradation by LAB phenolic acid decarboxylases. This negative effect of LAB phenolic acid decarboxylases could explain the result obtained in the present work (Figure 2), where TPC values after fermentation ([mg GAE/g_dw_]_FERM_) were lower than those of the corresponding unfermented samples ([mg GAE/g_dw_]_MW_). The reduction in TPC was significantly lower in samples pretreated at 4 W/g for 5 and 7 min, with reductions of approximately 12% and 19%, respectively. However, no clear correlation was observed between the reduction in TPC and the *L. plantarum* counts at the end of fermentation. Moreover, microbial enzymatic activity varies depending on the specific compound, with changes in TPC during fermentation resulting from the reduction in certain compounds, such as multi-glycosylated flavonoids and sinapoyl derivatives, and the increase in diglycosylated phenolics, including quercetin-3-O-diglucoside, kaempferol-3-O-diglucoside, and sinapic acid [52]. In addition, the different microwave treatments applied can affect the content of these specific compounds to varying extents, further influencing the overall changes in TPC observed during the fermentation process.

Based on the first part of this study, MW pretreatment at 4 W/g for 5 min was selected for further investigation due to its balance of energy efficiency, acceptable microbial growth, and minimal negative impact on phenolic content, both before and after 24 h fermentation with *Lactiplantibacillus plantarum*.

#### 3.1.2. Impact on Microstructure and FTIR-Spectra

Figure 3 shows the effects of microwave treatment on the residue microstructure. Cryo-FESEM micrographs of fresh and pasteurized broccoli stalk tissue before and after microwave treatment at 4 W/g for 5 min are shown. In the pasteurized sample, the cellular structures are still distinguishable, with well-defined parenchymatic cells and cell walls. The interior exhibits reticulated-like structures that result from water sublimation, suggesting the presence of an intracellular liquid fraction [53]. As observed, pasteurization could have caused certain permeabilization of structures but did not compromise the structural integrity of the cell walls, at least in the regions observed. Voids or empty intercellular spaces are also identified in the micrograph, with this being another sign of structure preservation. In contrast, the microwave-pretreated tissue exhibited pronounced structural degradation, characterized by less defined cellular structures and the formation of intracellular globular-like structures or aggregates. Micro-scale structural modifications in microwaved tissues were likely caused by the rapid and forceful evaporation of intracellular water (explosion phenomena at the hot spots), which disrupted cell walls and membranes and created channels that compromised cell integrity, potentially releasing cellular contents into the extracellular space [9]. Intercellular spaces were also identified in these samples but filled with liquid phase, which evidences the release of the internal liquid phase due to cell breakage. As for the rounded structures identified in the microwaved tissue, sub-protoplasts, endocytic vesicles, or globular-like structures may result from protoplast detachment and membrane invagination due to fast dehydration. These structures have been identified in cells and tissues subjected to high osmotic shocks during osmotic treatment [54,55], or in response to salinity or freezing [56]. These are mainly composed of plasma membrane material, which, although not always successfully, could potentially be incorporated into the protoplast in cases of rehydration. In the case of MW pretreated tissues, local water evaporation would be the origin of these structures’ formation. This level of tissue disruption might enhance the availability of nutrients, such as carbohydrates and nitrogenous compounds. Hot spot phenomena have been reported to promote the disruption of several plant tissues. Harahap et al. [57] evidenced that the hot spot effect induced by microwave irradiation facilitated the breakage of C-H and C-O bonds in bamboo, thus favoring cellulose conversion to bioethanol due to higher cellulose accessibility and lower substrate recalcitrance. Additionally, the breakdown of cell structures could facilitate the release of antioxidant molecules from their bound forms [11].

FTIR spectra of raw, pasteurized, and microwave samples are shown in Figure 4. The spectra exhibited similar profiles, differing only in absorbance across the spectral range.

The FTIR band attributed to the C=O stretching, C-C ring breathing vibrational mode, and C-O-H deformation mode of polysaccharides and pectin (∼1000 cm^−1^) slightly decreased in intensity following pasteurization and when lengthening the microwave treatment, particularly at 2 W/g. However, these bands intensified with microwave power, especially during treatments lasting 4–5 min. These changes may result from pasteurization and microwave processing breaking down or modifying cellulose, hemicellulose, and pectin.

The absorption features associated with amide-stretching bands of proteins in broccoli samples (∼1640 cm^−1^) remained unchanged following pasteurization and microwave treatment, suggesting minimal protein denaturation or aggregation from these thermophysical processes. Other authors reported no significant differences in total protein content between raw and cooked chickpea seeds, whether boiled, autoclaved, or microwaved [58]. However, a decrease in total sulfur-containing and aromatic amino acids was noted, while the total essential amino acids increased. A relatively low degree of protein denaturation was also observed in wheat germ after 20 min at 70 °C [59].

The absorption bands observed at approximately 1720 cm^−1^ and 1237 cm^−1^ correspond to the deformation vibrations of C=O and -OH groups in phenolic compounds. These bands, which were faint in all broccoli samples, decreased in intensity in pasteurized broccoli stalks compared to raw ones, despite the latter showing a statistically significant increase in total phenolic content. Following microwave treatment, the intensity of these bands varied, with treatments yielding higher total phenolic content not consistently corresponding to greater band intensities. Changes in phenolic compounds and antioxidants are also linked to variations in the O-H stretching region (3200–3600 cm^−1^), also affected by changes in the water content of samples. Although this region was not affected by pasteurization, notable changes were observed following microwave treatment, despite the replacement of the water lost through evaporation. In general, the intensity of this band increased with both the duration of microwave treatment and the rise in microwave power, likely due to a decrease in overall water content, the replacement of lost water, and, consequently, an increase in unbound water content.

As reported by Zdziobek et al. [46], a very narrow band can be observed in the spectra around 2323 cm^−1^ in fermented and fresh juice of broccoli. These authors associated this band with the C=N stretch vibration, possibly from glucosinolates or isothiocyanates. Nevertheless, this band was barely noticeable and remained unchanged in the present work for ground broccoli stalks following both pasteurization and microwave treatment.

The bands in the FTIR spectra around 2900 cm^−1^ correspond to the symmetric and asymmetric stretching vibrations of methyl (–CH_3_) and methylene (–CH_2_) groups, which are closely associated with lipids. The intensity of these bands decreased following pasteurization. Microwave treatment further reduced band intensity, with the bands nearly disappearing as processing time increased from 4 to 7 min at 2 W/g. However, at higher microwave powers, band intensity tended to increase when lengthening the treatment, suggesting that microwaves influence lipid composition and structure.

### 3.2. Comparative Study of Thermophysical Pretreatments

Ground broccoli stalks subjected to different thermophysical pretreatments (pasteurization, autoclaving, freezing/thawing, and microwaving at 4 W/g for 5 min) were analyzed for microbial growth, pH, and antioxidant properties at 0, 24, 48, 72, and 96 h of fermentation with *L. plantarum* CECT 749.

#### 3.2.1. Impact on Microbial Growth

Figure 5 shows changes in microbial counts and pH values during a 96 h fermentation period. A substantial microbial increase was observed within the first 24 h in all cases, rising from 7.3 ± 0.3 to 9.2 ± 0.2 log CFU/g. This trend aligns with previous findings that show that lactic acid bacteria thrive in cruciferous vegetables. Shokri et al. [60] also reported that the end point of the fermentation of broccoli puree made from broccoli florets and a mixed culture of *Leuconostoc mesenteroides* (BF1, BF2; 1:1) and *Lactobacillus plantarum* (B1) was reached after 24.1 h. However, this time was significantly reduced to 8.25 h and 9.9 h when thermal (60 °C for 7 min) or thermosonication (0.41 ± 0.02 W/g for 7 min at 60 °C) pretreatments were applied to the florets, respectively. In contrast, Iga-Buitrón et al. [24] reported longer fermentation times (2 days) to achieve maximum microbial growth during both spontaneous and induced fermentation using a co-culture of *Levilactobacillus brevis* (3M1) and *Lactococcus lactis* (3M8) strains in ground broccoli (including stems and florets) fermented in a 6% brine solution (14:9, *w*/*v*). Similarly, Salas-Millán et al. [52] observed a significant increase in LAB counts, reaching 8.04–8.50 log CFU/g, after three days of spontaneous fermentation of sliced broccoli stalks in a 6% brine solution (2:3, *w*/*v*), both undressed and dressed with garlic or mustard.

After the initial peak, microbial counts stabilized from 24 to 48 h, followed by a gradual decline likely due to nutrient depletion and increased acidity [30]. The corresponding pH drop from 6.04 ± 0.10 to 3.66 ± 0.06 in all samples supports this decline, as lactic and acetic acid production by lactic acid bacteria typically lowers the pH, creating a less favorable environment for bacterial proliferation [61]. The changes in pH depended on fermentation progression (*p*-value < 0.001), decreasing in all treatments until the second day of fermentation, without significant changes to the end of the experiment.

Interestingly, the freezing/thawing pretreatment resulted in reduced microbial counts and a less pronounced pH decrease compared to other pretreatments, consistently observed across all time points tested during fermentation. Since there were no significant differences in the RSC among the samples subjected to the different pretreatments (198 ± 9 mg GE/g_dw_), the observed differences in microbial activity may not be explained by sugar availability alone. One possible contributing factor is the formation of ice crystals during freezing, which may disrupt cellular integrity and potentially release compounds that inhibit or do not support fermentation effectively [62]. However, further analysis of metabolite profiles would be required to confirm this hypothesis. In contrast, pasteurization and microwaving resulted in both the highest microbial counts and the most significant pH decreases (*p*-value < 0.05), indicative of robust fermentation activity. Both treatments are known to preserve nutrients and bioactive compounds effectively, which may contribute to a more favorable environment for *L. plantarum* growth and acid production, corroborating findings from previous studies [63]. Furthermore, in the case of microwave-treated samples, the replacement of water lost through evaporation in its free form may have promoted microbial growth. Additionally, the impact of MW on the cellular structure would have facilitated microbial enzyme action and subsequent microbial growth, thus enhancing further fermentation efficiency. Autoclaved samples showed intermediate microbial counts and moderate pH changes, suggesting that high temperatures might degrade nutrients vital for bacterial growth and acid production.

#### 3.2.2. Impact on Antioxidant Properties

Figure 6 illustrates the evolution of the antioxidant properties of ground broccoli stalks during 96 h of fermentation with *Lactiplantibacillus plantarum*.

Pretreatments led to a significant increase (*p*-value < 0.05) in TPC as compared to the non-pretreated sample. This increase was particularly pronounced after MW treatment, resulting in a 2.2-fold increase, and it was almost negligible following the FR/TH treatment. These findings contrast with those of the preliminary study, where MW after pasteurization led to a decline in total phenolic content, limiting the enhancement relative to unprocessed tissue to only 1.1-fold. Since the efficiency of microwave heating is highly dependent on a material’s composition and is often hindered by non-uniform temperature distribution, even slight variations in shape, moisture content, or maturity level can significantly affect heating performance [17]. These inconsistencies make microwave pretreatment a challenging technology to implement at an industrial scale, where process uniformity is critical for ensuring consistent product quality.

The impact of thermophysical pretreatments on total flavonoids differed from their effect on total phenols. Samples subjected to FR/TH and AUTO pretreatments exhibited significantly higher flavonoid content (*p*-value < 0.05) compared to PAST and/or non-microwaved samples. Similarly, in Bas-Bellver et al. [64], AUTO pretreatment was the most effective in enhancing flavonoid content, likely due to the increased release of these compounds resulting from the combined effects of high pressure and heat on the tissue structure. For the same reason, autoclaving also resulted in the highest scavenging activity against DPPH and ABTS free radicals at the beginning of fermentation.

Fermentation of pretreated samples with *Lactiplantibacillus plantarum* for 24 h led to a slight but not statistically significant (*p*-value > 0.001) increase in TPC in MW and PAST samples, while a decrease was observed in AUTO and FR/TH samples. A significant overall decline in TPC was observed after 48 h of fermentation, stabilizing at constant values until the end of the process on day 4. This decline was particularly pronounced in MW samples, which eventually reached TPC levels comparable to those of the other treatments and similar to those found in fresh broccoli stalks. A similar reduction in TPC was reported by Salas-Millán et al. [52] in broccoli stalk slices, starting from day 3 of spontaneous fermentation in brine. The authors attributed the decrease to the diffusion of polyphenols from the broccoli slices into the surrounding liquid, driven by the increased cell wall permeability and structural breakdown induced by fermentation. Nevertheless, the same study revealed an increase in TPC on day 3 or 6 of fermentation in different dressed media. In this case, the increase was attributed to the structural degradation of the plant cell wall during fermentation, which may have facilitated the release and/or synthesis of new functional compounds, such as polyphenols. The transfer of intrinsic polyphenols from the dressing to the slices was also a contributing factor. The more pronounced increase and subsequent decrease in TPC observed in MW samples compared to PAST, AUTO or FR/TH samples could be attributed to microwave processing.

Findings from other studies further support the variability in TPC changes during fermentation. Iga-Buitrón et al. [24] reported that hydrolyzed phenol content remained stable until day 10 during both spontaneous and lactic acid-induced fermentation of broccoli stems and florets in brine, suggesting that microbial enzymatic activity had a limited effect on phenol degradation in that system. In contrast, Zdziobek et al. [46] observed a statistically significant increase in TPC from 4.17 ± 0.04 mg GAE/100 g to 30 ± 5 mg GAE/100 g and from 3.88 ± 0.14 mg GAE/100 g to 22 ± 2 mg GAE/100 g after an 8-day fermentation of broccoli florets juices with *Lactiplantibacillus plantarum* bacteria at 30 and 35 °C, respectively. Ye et al. [47] also found that lactic acid fermentation of autoclaved broccoli puree for 15 h at 30 °C resulted in a substantial increase in total phenolic acid concentration, ranging from 289 μg/g to between 903 μg/g and 3105 μg/g, depending on the bacterial strain used. As stated by Hou et al. [65], who observed an increase of 157% in the TPC of broccoli floret puree after 60 h of fermentation time with a mixture of *Lactiplantibacillus plantarum* and *Lactiplantibacillus pentosus*, the increase in TPC can be attributed to several factors, including the release of bound polyphenols, the breakdown of complex polyphenols into smaller, more bioavailable forms, and the activity of LAB enzymes such as cellulases and pectinases, which degrade cell wall polysaccharides. Additionally, LAB produce glycosidases, tannases, and esterases, which convert phenolic acid esters into aglycones and phenolic acids with higher antioxidant capacity. Microbial proteases may also contribute by dissociating protein–polyphenol complexes. Furthermore, the Folin–Ciocalteu method measures antioxidant capacity rather than direct TPC, and the lower pH during fermentation may enhance phenolic compound stability, further increasing TPC.

The total flavonoid content (TFC) in broccoli stalks behaved differently than TPC during fermentation with *L. plantarum*, despite being among the most abundant individual phenolics in raw broccoli stalks [52]. AUTO samples presented the highest TFC at all tested time points. Over the fermentation period, TFC in PAST samples increased from 0.18 ± 0.02 mg QE/g_dw_ to 0.27 ± 0.08 mg QE/g_dw_, with a sharp rise from day 2 to day 3. Similarly, during this interval, TFC in AUTO and MW samples showed a statistically significant increase (*p*-value < 0.05); however, the TFC levels in these samples at day 4 were identical to those of the corresponding unfermented samples. In the case of FR/TH samples, the TFC increase occurred later, between day 3 and day 4, but it was insufficient to compensate for the 58% loss registered, mainly during the first 24 h of fermentation. The reduction in TFC during broccoli stalk fermentation has been previously reported to be induced by the enzymatic activity of lactic acid bacteria, which caused an imbalance between the reduction in multi-glycosylated flavonoids and the increase in diglycosylated phenolics [52]. The increase in diglycosylated phenolics occurs later and to a greater extent than the reduction in multi-glycosylated flavonoids, which explains the temporary rise in TFC observed at certain points during fermentation. However, lactic acid bacteria may subsequently degrade these compounds through the activity of phenolic acid decarboxylase enzymes, leading to the final decline in TFC. From the different patterns observed in TFC changes during fermentation among samples, it can be inferred that the pretreatments have a different impact on the enzyme accessibility of lactic acid bacteria to specific compounds, directly related to the structural integrity of the vegetal tissue. In the case of PAST samples, the moderate fluctuations observed in TFC throughout fermentation suggest that pasteurization induced fewer alterations in the cellular structure of the ground broccoli stalks compared to the other thermophysical pretreatments, resulting in limited changes in the accessibility of flavonoid-related substrates for microbial enzymatic activity. This is consistent with the fact that pasteurization was applied to all samples before undergoing additional treatments such as MW, AUTO, or FR/TH.

The evolution of antioxidant capacity followed a similar pattern regardless of the analytical method employed, with values obtained via the ABTS method consistently higher than those from the DPPH method. This difference can be attributed to the distinct antioxidant mechanisms and sensitivities of each assay. The DPPH assay primarily follows a single electron transfer (SET) mechanism and is particularly responsive to lipophilic phenolic compounds and flavonoids with strong electron-donating groups. In contrast, the ABTS assay involves both SET and hydrogen atom transfer (HAT) mechanisms, allowing it to detect a broader spectrum of antioxidants, including both lipophilic and hydrophilic compounds such as ascorbic acid and certain peptides [24,66]. In all cases, fermentation led to a statistically significant decline (*p*-value < 0.05) in the ability of samples to scavenge ABTS and DPPH free radicals, and this decline was not influenced by the pretreatment applied before fermentation. Consequently, AUTO samples, which initially exhibited the highest antioxidant activity, maintained this status throughout the fermentation process. Regardless of the pretreatment, antioxidant capacity measured by the DPPH method reached its lowest point after 48 h of fermentation, showing a 68 ± 6% decrease compared to unfermented counterparts. However, from 48 h onward, this reduction lessened to 40 ± 5% and remained stable until the end of the process. Similarly, antioxidant capacity measured by the ABTS method reached its lowest value after 72 h of fermentation, with a 15 ± 5% decrease compared to unfermented samples. The values registered at this stage remained stable until the completion of 96 h of fermentation. These results contrast with the findings of Iga-Buitrón et al. [24], Hou et al. [65], and Zdziobek et al. [46], who reported a significant increase in the ability to reduce free radicals during lactic acid bacteria fermentation in various broccoli-based products.

While a specific analysis of the limitations of this study was not conducted, they can be broadly attributed to factors such as the characteristics and variability of the raw materials, the experimental design, the analytical methods employed, and the equipment used.

## 4. Conclusions

Overall, this study demonstrates that pretreatments can significantly influence the success of fermentation in terms of both microbial development and the retention or enhancement of bioactive compounds. Microwaving at 4 W/g for 5 min following pasteurization appears to be the most favorable option to obtain a fermented product with the highest phenolic content and microbial viability. In contrast, autoclaving is more effective when the goal is to maximize flavonoid concentration and antioxidant activity.

Although 96 h fermentation with *Lactiplantibacillus plantarum* was not proved to consistently enhance the antioxidant properties of ground broccoli stalks, it is expected that further research focused on determining specific bioactive compounds or other fermentation products (phenolics, volatile compounds) will reveal valuable information about the success of fermentation after applying thermophysical pretreatments. It would also be interesting to explore the changes in the bioavailability of specific bioactive compounds as a result of the treatments applied to the plant matrix.

Maximizing the functional and nutritional qualities of fermented vegetable by-products can effectively contribute to food systems circularity by the valorization of discards and vegetable wastes. Therefore, the findings of the present research provide valuable insights towards the development of sustainable food systems.

## Figures and Tables

**Figure 1 foods-14-01809-f001:**
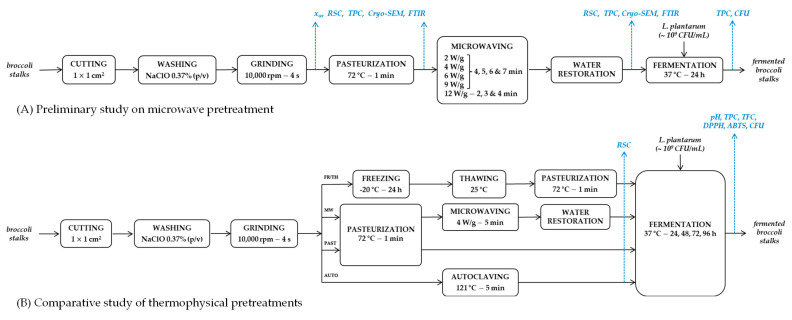
Flow diagrams including the analytical determinations performed during (**A**) the preliminary study on microwave pretreatment and (**B**) the comparative study of thermophysical pretreatments.

**Figure 2 foods-14-01809-f002:**
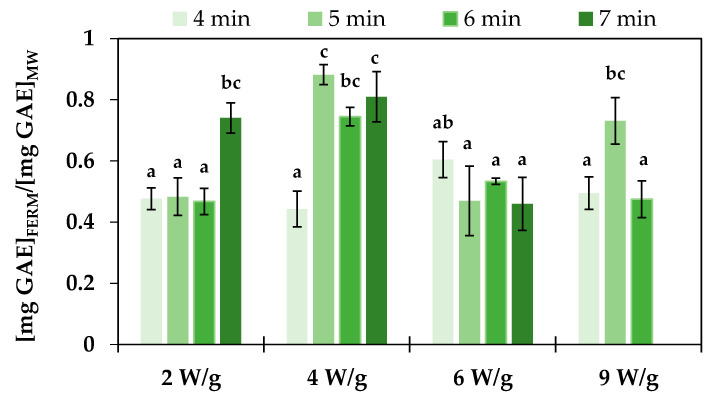
Effect of microwave treatments (power and duration) on changes in total phenolic content in pasteurized ground broccoli stalks after 24 h of fermentation with *Lactiplantibacillus plantarum*. Error bars represent the standard deviation of the means (n = 3). Different letters (a–c) indicate significant differences among means, determined by Tukey’s Honest Significant Difference test at a 95% confidence level.

**Figure 3 foods-14-01809-f003:**
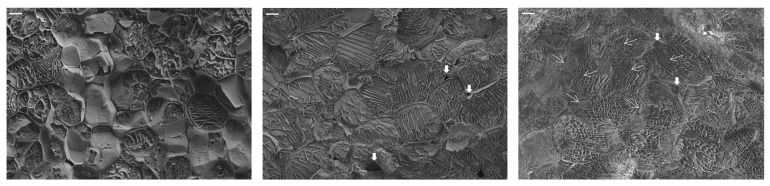
Cryo-FESEM micrographs of ground fresh broccoli stalks (**left**), ground pasteurized broccoli stalks (**center**), and ground broccoli stalks subjected to microwave treatment (4 W/g for 5 min) (**right**), observed at 300× of magnification (bar = 20 microns). Coarse white arrows indicate empty intercellular spaces (voids) in the pasteurized tissue and intercellular spaces filled with liquid phase in the microwaved-treated tissue. Thin arrows indicate sub-protoplasts, vesicles, or globular plasma membrane structures in the microwaved sample.

**Figure 4 foods-14-01809-f004:**
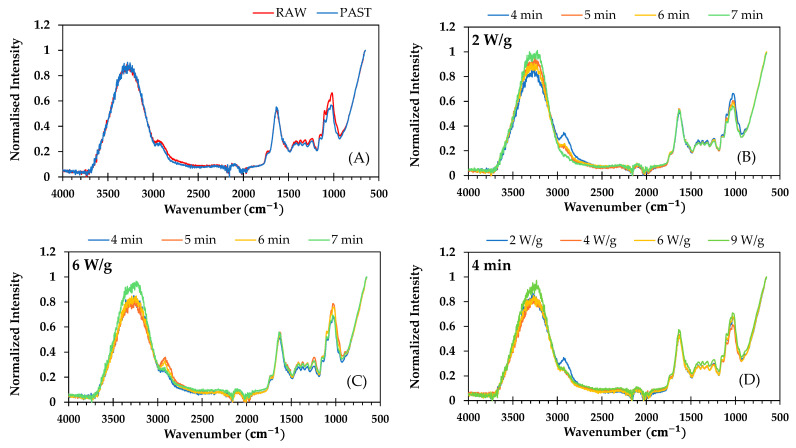
FTIR spectra of (**A**) fresh (RAW) and pasteurized (PAST) ground broccoli stalks; (**B**,**C**) pasteurized ground broccoli stalks subjected to microwave treatment at constant power of 2 and 6 W/g, respectively, with varying durations (4, 5, 6, and 7 min); and (**D**) pasteurized ground broccoli stalks treated at different powers (2, 4, 6, and 9 W/g) for 4 min.

**Figure 5 foods-14-01809-f005:**
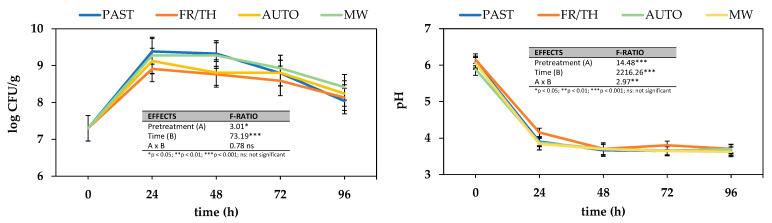
Effect of different thermophysical pretreatments on *Lactiplantibacillus plantarum* counts and pH depletion during 96 h fermentation. Error bars represent the standard deviation of the means (n = 3). Pretreatments: PAST = pasteurization, FR/TH = freeze–thawing, AUTO = autoclaving, MW = microwaving.

**Figure 6 foods-14-01809-f006:**
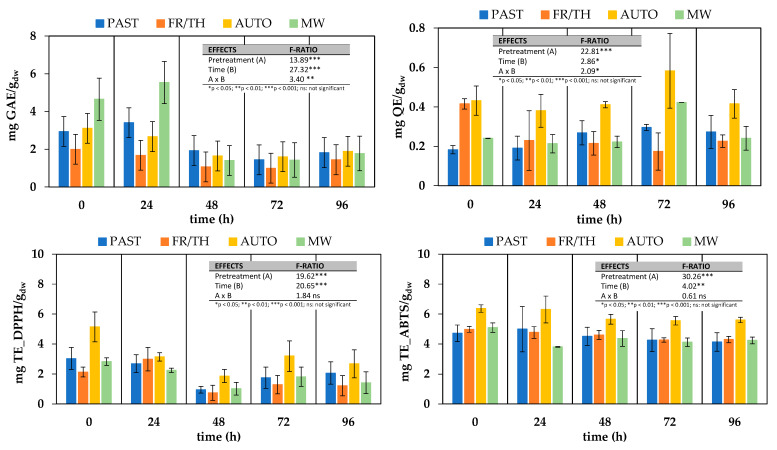
Effect of different thermophysical pretreatments on antioxidant properties of ground broccoli stems during 96 h fermentation with *L. plantarum*. Error bars indicate the standard deviation (n = 3). Antioxidant properties include total phenols (mg GAE/g_dw_), total flavonoids (mg QE/g_dw_), and DPPH and ABTS scavenging ability (mg TE/g_dw_). Pretreatments: PAST = pasteurization, FR/TH = freeze–thawing, AUTO = autoclaving, MW = microwaving.

**Table 1 foods-14-01809-t001:** Effect of microwave power and duration on reducing sugar content (RSC), total phenolic content (TPC), and microbial population after 24 h of fermentation of pasteurized broccoli ground stalks. Data are presented as mean ± standard deviation of three replicates. NF: not undergoing fermentation due to browning or calcination phenomena in microwave-treated samples.

TREATMENT		RSC (mg GE/g_dw_)	TPC (mg GAE/g_dw_)	*L. plantarum* (Log CFU/g)
RAW		144 ± 2 ^ab^	1.97 ± 0.06 ^abc^	NF
PASTEURIZED		195 ± 5 ^f^	2.59 ± 0.08 ^cde^	NF
2 W/g	4′	193.2 ± 1.3 ^ef^	2.55 ± 0.02 ^bcd^	9.46 ± 0.05 ^ab^
	5′	180.7 ± 1.2 ^def^	2.32 ± 0.10 ^abcd^	9.29 ± 0.02 ^ab^
	6′	171.7 ± 1.1 ^bcdef^	2.56 ± 0.02 ^bcd^	9.23 ± 0.09 ^ab^
	7’	147.3 ± 1.4 ^abc^	2.1 ± 0.3 ^abc^	9.044 ± 0.003 ^a^
4 W/g	4′	167 ± 3 ^bcde^	1.795 ± 0.008 ^ab^	9.0 ± 0.2 ^a^
	5′	147.7 ± 0.9 ^abc^	2.31 ± 0.13 ^abcd^	9.48 ± 0.13 ^ab^
	6′	146 ± 2 ^ab^	1.91 ± 0.04 ^abc^	9.46 ± 0.11 ^ab^
	7′	177 ± 2 ^cdef^	1.55 ± 0.04 ^a^	9.70 ± 0.08 ^bc^
6 W/g	4′	166 ± 7 ^bcde^	2.62 ± 0.11 ^bcd^	9.21 ± 0.10 ^ab^
	5′	156 ± 5 ^abcd^	2.7 ± 0.3 ^bcd^	9.22 ± 0.12 ^ab^
	6′	147 ± 4 ^ab^	2.18 ± 0.11 ^abcd^	9.28 ± 0.05 ^ab^
	7′	156 ± 8 ^abcd^	3.02 ± 0.04 ^d^	9.23 ± 0.02 ^ab^
9 W/g	4′	171 ± 2 ^bcdef^	2.61 ± 0.13 ^bcd^	10.13 ± 0.14 ^c^
	5′	180 ± 3 ^def^	2.6 ± 0.2 ^bcd^	10.16 ± 0.12 ^c^
	6′	133 ± 9 ^a^	2.7 ± 0.2 ^cd^	10.2 ± 0.3 ^c^
	7′	-	3.0 ± 0.2 ^d^	NF
12 W/g	2′	-	2.37 ± 0.11 ^bcd^	NF
	3′	-	2.25 ± 0.02 ^bcd^	NF
	4′	-	3.40 ± 0.10 ^f^	NF

Different superscripts (a–f) within the same column indicate significant differences among means, determined by Tukey’s Honest Significant Difference test at the 95% confidence level.

## Data Availability

The data supporting the findings of this study are available from the corresponding author upon reasonable request.

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
