# Peer review of "Assessing the Impact of Microwaves and Other Disruptive Pretreatments on Lactiplantibacillus plantarum Growth and the Antioxidant Properties of Broccoli Stalks"

_foods, 2025, doi:10.3390/foods14101809_

Round 1
Reviewer 1 Report
Comments and Suggestions for Authors
Dear Editor of Food, MDPI, thank you for considering me as part of the review group for your prestigious journal. I have carefully reviewed the manuscript entitled "Assessing the impact of microwaves and other disruptive Pretreatments on Lactiplantibacillus plantarum growth and Antioxidant properties of Broccoli stalks" and I consider it suitable for publication after the authors make the following adjustments or respond to each comment.
Comment to authors
- Line 17-18 should be modified according to how they appear in the results and methodology, first Cryo-SEM and then FTIR.
- At the end of the abstract you should give a general conclusion about these technologies for food waste and their use in the industry.
- To further support this work, the introduction includes the concept of a zero-waste economy that aims to redesign production and consumption to eliminate waste, encouraging reuse, recycling, and circular resource flows for a more sustainable future. Cite to: (2022). Trends in Sustainable Green Synthesis of Silver Nanoparticles Using Agri‐Food Waste Extracts and Their Applications in Health. Journal of Nanomaterials, 2022(1), 8874003.
- The methodology must include a Figure that exemplifies how the microwave design was carried out followed by fermentation (2.2 and 2.3).
- In the results section, where it refers to FTIR, the word "peak" should be replaced with "band." Bands is more appropriate for this determination, as each signal has its own length and width.
- In the results section on antioxidant activity, each antioxidant mechanism (SET and HAT) present in each technique (ABTS and DPPH) needs further explanation, as well as which one predominates and why, based on the sample components.
Author Response
ANSWERS TO REVIEWER 1 COMMENTS:
Comment 1. Line 17-18 should be modified according to how they appear in the results and methodology, first Cryo-SEM and then FTIR.
Response 1. This change has been made as suggested
Comment 2. At the end of the abstract, you should give a general conclusion about these technologies for food waste and their use in the industry.
Response 2. As suggested, a general conclusion regarding the suitability of microwaving and other disruptive methodologies for treating pasteurized broccoli stalks prior to fermentation has been added at the end of the Abstract. As the addition caused the abstract to exceed the 200-word limit set by the journal, the authors have revised and shortened it accordingly.
Comment 3. To further support this work, the introduction includes the concept of a zero-waste economy that aims to redesign production and consumption to eliminate waste, encouraging reuse, recycling, and circular resource flows for a more sustainable future. Cite to: (2022). Trends in Sustainable Green Synthesis of Silver Nanoparticles Using Agri‐Food Waste Extracts and Their Applications in Health. Journal of Nanomaterials, 2022(1), 8874003.
Response 3. The authors appreciate the reviewer’s suggestion to include the reference addressing the concept of a zero-waste economy, which indeed aligns with the general topic of by-product valorization and sustainable resource use. However, after careful consideration, the authors believe that the current set of references sufficiently supports the discussion around circularity and sustainability within the context of broccoli stalk valorization. To maintain focus and conciseness in the manuscript, no additional citations have been included at this point. Nevertheless, the authors acknowledge the relevance of the suggested reference and would be pleased to incorporate it should the editor consider it essential for the completeness of the manuscript.
Comment 4. The methodology must include a Figure that exemplifies how the microwave design was carried out followed by fermentation (2.2 and 2.3).
Response 4. In line with the reviewer’s recommendation, a new figure illustrating the sequence of unit operations and the analytical determinations performed during both the preliminary study on microwave pretreatment and the comparative study of thermophysical pretreatments has been added to the Materials and Methods section of the revised manuscript. The numbering of the existing figures has been updated accordingly to reflect this new incorporation.
Comment 5. In the results section, where it refers to FTIR, the word "peak" should be replaced with "band." Bands is more appropriate for this determination, as each signal has its own length and width.
Response 5. The authors thank the reviewer for this helpful observation. In the revised version of the manuscript, the term "peak" has been replaced with "band" in the FTIR results section, as suggested, to more accurately reflect the nature of the spectral signals.
Comment 6. In the results section on antioxidant activity, each antioxidant mechanism (SET and HAT) present in each technique (ABTS and DPPH) needs further explanation, as well as which one predominates and why, based on the sample components.
Response 6. The authors appreciate the reviewer’s valuable suggestion. In response, the revised manuscript includes a more detailed explanation of the antioxidant mechanisms involved in each assay. The DPPH method primarily follows a single electron transfer (ET) mechanism and is more responsive to lipophilic phenolic compounds and flavonoids with strong electron-donating groups. In contrast, the ABTS assay operates through both ET and hydrogen atom transfer (HAT) mechanisms, making it more versatile and capable of detecting a wider range of antioxidants, including hydrophilic compounds such as ascorbic acid and certain peptides. Given the composition of broccoli stalks, rich in phenolics, flavonoids, and possibly hydrophilic antioxidants, ABTS may provide a broader antioxidant profile, while DPPH mainly reflects the activity of specific phenolic components. These considerations have been addressed in the revised results section to better contextualize the observed antioxidant activity, and a new reference has been introduced to support this explanation (doi: 10.1016/j.jff.2015.01.047).

Reviewer 2 Report
Comments and Suggestions for Authors
Dear authors, below are some considerations with my suggestions for improving the work and reconsidering accepting the paper for publication.
Limitations of the study: please revise and include in the paper the limitations in the discussion and conclusion of the results (if not performed during the study include this as a limitation)
- Authors did not analyzed specific bioactive compounds or bioaccessibility assays? This is an important gap, please discuss it.
- A single-strain was utilized (L. plantarum); how about other probiotic strains or co-cultures?
- Please better explain the method for water restoration post-microwaving
- Please better explain the replication method in MM
- Samples with browning did not undergo fermentation, how this affected analyses?
- Did authors utilized a non-pasteurized control?
- Autoclaving and microwaving were applied post-pasteurization? How about the individual treatment impacts?
- A multivariate analysis to correlate pretreatment conditions with antioxidant outcomes was considered to be performed?
- The study does not assess the sensory attributes or shelf-life testing of the fermented products?
Keywords: please improve this section, avoid using words that were already utilized in the title
MM: please clarify power-to-mass ratios and duration/power selections. Time-treatment interactions were modelled as fixed or random effects?
Results and discussion: link findings to industrial applications. Standard deviations in antioxidant measurements are occasionally high, how the implications of variability affect it?
Numbers after the comma needs to be standardized
Tables and figures are not as MDPI’s formatting standards. Screenshots or low-resolution images of tables/charts are included, which lack professional polish and hinder readability.
Interpretation attributing microbial growth patterns to substrate accessibility without additional information is speculative.
Conclusion: please include the future research directions and practical applications. Can authors suggest practical recommendations for food systems based on findings?
Author Response
ANSWERS TO REVIEWER 2 COMMENTS:
Comment 1. Limitations of the study: please revise and include in the paper the limitations in the discussion and conclusion of the results (if not performed during the study include this as a limitation).
Response 1.
Thank you very much for your suggestion. We acknowledge that a specific analysis of the limitations was not performed when conducting this research. We will consider this in our future work. Following this reviewer recommendation, the following sentence has been added at the end of the Discussion: “While a specific analysis of the limitations of this study was not conducted, they can be broadly attributed to factors such as the characteristics and variability of the raw materials, the experimental design, the analytical methods employed, and the equipment used”.
Comment 2. Authors did not analyzed specific bioactive compounds or bioaccessibility assays? This is an important gap, please discuss it.
Response 2. We thank the reviewer for this valuable observation and fully agree that the analysis of specific bioactive compounds and bioaccessibility assays represents an important issue for further research. As noted in the Conclusions section of the manuscript, " it is expected that further research focused on determining specific bioactive compounds or other fermentation products (phenolics, volatile compounds) will reveal valuable information about the success of fermentation after applying thermophysical pretreatments. It would also be interesting to explore the changes in the bioavailability of specific bioactive compounds as a result of the treatments applied to the plant matrix”. While the current study did not include these specific analyses, the experimental proposed was an important step in establishing the microwave treatment conditions for pasteurized broccoli stalks (4 W/g for 5 minutes) before fermentation with Lactiplantibacillus plantarum. It also allowed the authors to determine the fermentation duration that, for each of the disruptive pretreatments tested, led to greater microbial growth and fermented products with the most favourable antioxidant properties. With these key parameters now defined, follow-up studies will be conducted to evaluate the impact of the pretreatments on specific bioactive compounds and their bioavailability, along with other functional and compositional attributes.
Comment 3. A single strain was utilized (L. plantarum), how about other probiotic strains or co-cultures?
Response 3. As stated in the Materials and Methods section, “L. plantarum CECT 749 was selected due to its probiotic potential, its Qualified Presumption of Safety (QPS) status, and its adaptability to diverse ecological environments”. The choice was also supported by our previous findings, in which L. plantarum CECT 749 outperformed other probiotic strains, such as L. reuteri CECT 925 and L. salivarius spp. salivarius CECT 4063, achieving the highest microbial counts in both ground and chopped broccoli stems after 24 hours of fermentation (doi: 10.3934/microbiol.2024013). Accordingly, the citation to this previous work has been introduced earlier in the manuscript, within the Materials and Methods section, to clarify the basis for strain selection. As a result, the numbering of the subsequent references has been updated to reflect this change. We agree that exploring co-cultures and other probiotic strains is a valuable direction for future research, particularly now that the suitability of applying disruptive treatments prior to fermentation has been established.
Comment 4. Please better explain the method for water restoration post-microwaving.
Response 4. As stated in the Materials and Methods section, the amount of water to be added after microwaving was calculated based on the weight loss observed in the samples during the microwaving and subsequent cooling steps. To prevent contamination with microbial strains other than the one selected for this study, the water was sterilized prior to use. It was then carefully reintegrated into the plant tissue using a sterile spatula to ensure even distribution. This additional methodological detail has been incorporated into the revised manuscript.
Comment 5. Please better explain the replication method in MM.
Response 5. The authors apologize for not having properly explain the replication strategy. In response, the Statistical Analysis subsection of the Materials and Methods has been expanded to provide greater detail. Specifically, each combination of microwave power and time, and each disruptive pretreatment, was applied in three independent replicates. Fermentation of each pretreated sample with Lactiplantibacillus plantarum was likewise carried out independently in three separate glass jars. Where applicable, analytical measurements were performed in duplicate on aliquots taken from each jar. Thus, as indicated in the Statistical Analysis subsection, all reported values derived from three independent experimental units with two technical replicates each, and results were expressed as mean ± standard deviation.
Comment 6. Samples with browning did not undergo fermentation, how this affected analyses?
Response 6.
Browning reveals a remarkable incidence of Maillard reactions, hydrothermal degradation of sugars and potential generation of antimicrobial compounds such furfural and hydroxymethyl furfural. According to previous research, these samples were not considered appropriate for further fermentation; nevertheless, properties of the microwaved residue were analysed and compared with the rest of treatments.
This result did not affect analyses, since samples removed corresponded to a particular power/time combination as detailed in Table 1 (9 W/g 7 min, and all 12 W/g). In these cases, it is indicated “NF: not undergoing fermentation, due to browning or calcination phenomena in microwave-treated samples.”
After reading this reviewer’s comment, we believe that maybe it has been understood that particular samples presenting browning were removed from the analysis, but it was the whole batch of samples corresponding to a particular power/time combination which did not undergo further fermentation. We have tried to clarify this in the text to avoid misunderstanding “microwave treatments causing calcinations or significant browning were not considered for further fermentation (NF in table 2).”
Comment 7. Did authors utilize a non-pasteurized control?
Response 7. A non-pasteurized control was not included in this study. Pasteurization was applied to all samples, except those subjected to autoclaving, with the primary objective of inactivating the native microbial population. This step was necessary to ensure that the inoculated Lactiplantibacillus plantarum strain was the only microorganism actively growing during fermentation, allowing for a controlled assessment of the effects of pretreatment on microbial growth and antioxidant outcomes.
Comment 8. Autoclaving and microwaving were applied post-pasteurization? How about the individual treatment impacts?
Response 8. As clarified in the newly added Figure 1 of the revised manuscript, autoclaving was applied individually to the sanitized ground broccoli stalks prior to fermentation. In the case of microwave treatments, a pasteurization step was performed beforehand. This precaution was taken because the effectiveness of microwaving (particularly at lower power levels and shorter durations) on inactivating the natural microbial population was uncertain. Therefore, pasteurization was used to ensure that the inoculated Lactiplantibacillus plantarum strain would dominate the fermentation process.
Comment 9. A multivariate analysis to correlate pretreatment conditions with antioxidant outcomes was considered to be performed?
Response 9. We appreciate the reviewer’s suggestion. A multivariate analysis was considered; however, it was not performed in this study. Instead, a multifactor ANOVA was conducted to evaluate the impact of each pretreatment condition (FR/TH, AUTO, PAST and MW) and fermentation time (0, 24, 48, 72 and 96 h) on the antioxidant outcomes. The F-ratios and p-values for each individual factor and their interactions are presented in the corresponding figures of the original manuscript.
Comment 10. The study does not assess the sensory attributes or shelf-life testing of the fermented products.
Response 10. Sensory attributes and shelf-life testing were not within the scope of the present study. In fact, the fermented product is not intended for direct consumption, but rather for further stabilization and transformation into a powdered ingredient for use in functional food formulations.
Comment 11. Keywords: please improve this section, avoid using words that were already utilized in the title.
Response 11. Authors have revised the keywords to avoid repetition of terms already included in the title, aiming to enhance the discoverability of the manuscript. The updated keywords are now more specific and complementary to the title content: broccoli residues, plant-based fermentation, LAB fermentation, thermophysical pretreatments, probiotic.
Comment 12. MM: please clarify power-to-mass ratios and duration/power selections. Time-treatment interactions, were modelled as fixed or random effects?
Response 12. Ratios and duration were based on previous experiences of the group on microwave treatments applied to intensify biological processes (doi: 10.1016/j.fbp.2016.07.001), together with preliminary tests which allowed to discard treatments evidencing thermal degradation of the plant matrix and its compounds (calcination or browning phenomena).
Comment 13. Results and discussion: link findings to industrial applications. Standard deviations in antioxidant measurements are occasionally high, how the implications of variability affect it?
Response 13. The authors appreciate this reviewer comment. We would like to note, however, that this piece of research was focused on understanding the effects of microwave and other pretreatments on further fermentation, and at the moment is far from industrial development. Regarding the variability in the antioxidant measurements, we understand that it can be considered high, but this result is rather common in this kind of matrices when using spectrophotometric methods for antioxidant properties analysis. That is one of the reasons why the authors propose, as part of the Conclusions, to continue the study using more specific analytical techniques to identify specific phenolic constituents or volatile compounds.
Comment 14. Numbers after the comma needs to be standardized.
Response 14. Thank you very much for your recommendation. We agree that researchers may use the same number of decimals in tables to indicate variability of results. However, we are more familiar with the use of theory of error for significant figures, which is a ground and extensively used theory. The theory of error attributes to the mean the number of significant figures determined by its standard deviation. Briefly, theory of error for expressing uncertainty when repeated measurements are performed states de following:
“When determining the mean and standard deviation based on repeated measurements:
- The mean cannot be more accurate than the original measurements.
- The standard deviation provides a measurement of experimental uncertainty.
- Experimental uncertainty (standard deviation) should be rounded to one significant figure (first value different to zero). The only exception is when the uncertainty has a leading digit of 1, when a second digit should be kept”.
Therefore, result of each repeated measurement has its own number of significant figures in the mean and standard deviation values, which are given by the first digit different from zero in the standard deviation (except for the first figure being 1).
[This information can be checked, for instance, in: Measurement and Uncertainty Analysis Guide. (https://users.physics.unc.edu/~deardorf/uncertainty/UNCguide.pdf), or Introduction to measurements and error analysis (chrome-extension://efaidnbmnnnibpcajpcglclefindmkaj/https://astro.pas.rochester.edu/~aquillen/phy141/lectures/pdfs/uncertainty_nc.pdf)]
As a result of this method, not all values are reported with the same number of decimal places. However, this approach is both rigorous and scientifically valid, as it ensures that the reported values reflect their precision and uncertainty appropriately. Given that this criterion is well-established and valid, and consistent with best practices in scientific reporting, the authors regret not making the change suggested by the reviewer.
Comment 15. Tables and figures are not as MDPI’s formatting standards. Screenshots or low-resolution images of tables/charts are included, which lack professional polish and hinder readability.
Response 15. Authors apologize for the inconvenience caused. The high-quality files are available for the editorial team to incorporate during the production process. This measure aims to enhance the clarity, professionalism, and readability of the visual elements. The reviewer’s feedback is appreciated and has contributed to improving the presentation of the work.
Comment 16. Interpretation attributing microbial growth patterns to substrate accessibility without additional information is speculative.
Response 16. Our discussion is based on the observed results and on other authors’ findings. In some cases, discussion may include some hypothesis which could have resulted speculative, as deduced from this reviewer’s comments. Following this reviewer suggestion, we have tried to reduce speculative information and be clear when discussing an hypothesis.
Comment 17. Conclusion: please include the future research directions and practical applications. Can authors suggest practical recommendations for food systems based on findings?
Response 17. Conclusions have been modified according to this and other reviewers.

Reviewer 3 Report
Comments and Suggestions for Authors
The extraction of bioactive compounds from food waste is of great significance. Thermophysical pretreatments and fermentation can improve the efficiency of extraction, but have positive or negative influences on the antioxidant activity of bioactive compounds. In this manuscript, ground broccoli stalks are selected as plant waste, and are subjected to several thermophysical pretreatments (pasteurization, autoclaving, freezing/thawing, and microwaving); the influences of thermophysical pretreatments on total reducing sugars content, total phenolic content, and Lactobacillus plantarum CECT 749 ability to grow on ground broccoli stalks are discussed. In addition, microscopic observations and FTIR spectra are evaluated for the explanation of the relative changes.
This investigation accords with the scope of Foods - Special issue: Functional Ingredients from Food Waste and By-Products. Overall, this manuscript is clearly organized, and the important points are presented. The materials and methods are also clearly described. The positive and interesting results are obtained. After minor revision, this manuscript can be accepted for publishing.
Please note the following minor issues:
Introduction Section:
When FAO appears for the first time, its full name (Food and Agriculture Organization of the United Nations) should be given.
Results and Discussion Section:
Lines 294-295: “The RSC values for the raw material were slightly higher than those reported by Rosa et al” should be replaced by “The RSC value for the raw material was slightly higher than that ---”.
Lines 300-301: “Pasteurization resulted in a 35.4% increase in RSC and a 23.3% rise in TPC compared to raw broccoli stalks”. The increase percentage (23.3%) of TPC is not correct according to Table 1. The correct results are as follows:
For RSC, 100%*(195-144)/144 = 35.4%
For TPC, 100%*(2.59-1.97)/1.97 = 31.5%
Line 430: “the effects of microwave treatment in the residue microstructure”: “in” should be replaced by “on”.
References Section:
The case sensitivity of paper titles in references should be consistent. For example: “Effect of different cooking methods on structure and quality of industrially frozen carrots”, “Antioxidants Bound to an Insoluble Food Matrix: Their Analysis, Regeneration Behavior, and Physiological Importance”
- Paciulli, M.; Ganino, T.; Carini, E.; Pellegrini, N.; Pugliese, A.; Chiavaro, E. Effect of different cooking methods on structure and quality of industrially frozen carrots. J. Food Sci. Technol. 2016, 53, 2443-2451.
- Cömert, E.D.; Gökmen, V. Antioxidants Bound to an Insoluble Food Matrix: Their Analysis, Regeneration Behavior, and Physiological Importance. Compr. Rev. Food Sci. Food Saf. 2017, 16(3), 382-399.
Author Response
ANSWERS TO REVIEWER 3 COMMENTS:
Comment 1. When FAO appears for the first time, its full name (Food and Agriculture Organization of the United Nations) should be given.
Response 1. In accordance with Reviewer 3’s suggestion, the full name (Food and Agriculture Organization of the United Nations) has been provided at its first mention in the revised manuscript, followed by the abbreviation (FAO).
Comment 2. Lines 294-295: “The RSC values for the raw material were slightly higher than those reported by Rosa et al” should be replaced by “The RSC value for the raw material was slightly higher than that ---”.
Response 2. Authors appreciate the Reviewer 3’s careful observation. The sentence has been revised accordingly to ensure grammatical accuracy and clarity.
Comment 3. Lines 300-301: “Pasteurization resulted in a 35.4% increase in RSC and a 23.3% rise in TPC compared to raw broccoli stalks”. The increase percentage (23.3%) of TPC is not correct according to Table 1. The correct results are as follows:
For RSC, 100%*(195-144)/144 = 35.4%
For TPC, 100%*(2.59-1.97)/1.97 = 31.5%
Response 3. The authors apologize for the oversight and thank Reviewer 3 for carefully pointing out this detail. The percentage increase in TPC has been corrected in the revised manuscript to accurately reflect the value derived from the data presented in Table 1.
Comment 4. Line 430: “the effects of microwave treatment in the residue microstructure”: “in” should be replaced by “on”.
Response 4. Authors appreciate the Reviewer 3’s careful suggestion. The sentence has been revised, and "in" has been replaced with "on" as recommended.
Comment 5. The case sensitivity of paper titles in references should be consistent. For example: “Effect of different cooking methods on structure and quality of industrially frozen carrots”, “Antioxidants Bound to an Insoluble Food Matrix: Their Analysis, Regeneration Behavior, and Physiological Importance”.
Paciulli, M.; Ganino, T.; Carini, E.; Pellegrini, N.; Pugliese, A.; Chiavaro, E. Effect of different cooking methods on structure and quality of industrially frozen carrots. J. Food Sci. Technol. 2016, 53, 2443-2451.
Cömert, E.D.; Gökmen, V. Antioxidants Bound to an Insoluble Food Matrix: Their Analysis, Regeneration Behavior, and Physiological Importance. Compr. Rev. Food Sci. Food Saf. 2017, 16(3), 382-399.
Response 5. In response to this comment, the authors have carefully revised the manuscript to ensure greater consistency and homogeneity in the capitalization of paper titles in the reference list. Authors appreciate the reviewer’s attention to this detail, which has helped improve the overall presentation of the manuscript.

Reviewer 4 Report
Comments and Suggestions for Authors
This study evaluated the impact of microwave pretreatment on fermentation of pasteurized ground broccoli stalks with Lactiplantibacillus plantarum by determining reducing sugars, total phenolics, and microbial growth were analyzed, along with FTIR spectra and SEM microstructure. The research may contribute valuable insight into the selection of appropriate pretreatments for maximizing the functional and nutritional qualities of fermented vegetable by-products. However, revisions should be conducted before acceptance.
1 line 153-155, did the microwave conditions including microwave power and processing time being chosen by your preliminary test results or according to references?
2 table 1 is not normative, the three-wire meter is suggested.
3 the discussions are not convictive or strict, for example, line 361-362, TPC can react with reducing sugars? And why sugar was degraded during microwave heating?
4 the conclusion section is too long, and the concise and accurate conclusion should be provided. Please reorganize this section.
Author Response
ANSWERS TO REVIEWER 4 COMMENTS:
Comment 1. Line 153-155, did the microwave conditions including microwave power and processing time being chosen by your preliminary test results or according to references?
Response 1. Microwave power and exposure times were selected on the basis of both preliminary trials and relevant literature (doi: 10.1016/j.fbp.2016.07.001), then adapted to the specifications of the laboratory microwave oven. Five power-to-mass ratios were initially evaluated: 2 W/g (100 W/50 g), 4 W/g (300 W/75 g), 6 W/g (300 W/50 g), 9 W/g (450 W/50 g), and 12 W/g (600 W/50 g), each applied for 4, 5, 6, and 7 min. Partial browning was observed at 9 W/g after 7 min and at all 12 W/g conditions. Consequently, the final set of microwave pre-treatments comprised 2, 4, 6 and 9 W/g for 4, 5, 6, and 7 min, and 12 W/g for 2, 3, and 4 min. This selection ensured a broad yet non-browning range of energy inputs for subsequent analyses, while also highlighting the limitations encountered at the highest power level. A brief explanation of how the microwave conditions were selected has been added to the revised manuscript.
Comment 2. Table 1 is not normative, the three-wire meter is suggested.
Response 2. We appreciate the feedback regarding Table 1 and the suggestion of the three-wire meter. However, we are more familiar with the use of theory of error for significant figures, which is a ground and extensively used theory. The theory of error attributes to the mean the number of significant figures determined by its standard deviation. Briefly, theory of error for expressing uncertainty when repeated measurements are performed states de following:
“When determining the mean and standard deviation based on repeated measurements:
- The mean cannot be more accurate than the original measurements.
- The standard deviation provides a measurement of experimental uncertainty.
- Experimental uncertainty (standard deviation) should be rounded to one significant figure (first value different to zero). The only exception is when the uncertainty has a leading digit of 1, when a second digit should be kept”.
Therefore, result of each repeated measurement has its own number of significant figures in the mean and standard deviation values, which are given by the first digit different from zero in the standard deviation (except for the first figure being 1).
[This information can be checked, for instance, in: Measurement and Uncertainty Analysis Guide. (https://users.physics.unc.edu/~deardorf/uncertainty/UNCguide.pdf), or Introduction to measurements and error analysis (chrome-extension://efaidnbmnnnibpcajpcglclefindmkaj/https://astro.pas.rochester.edu/~aquillen/phy141/lectures/pdfs/uncertainty_nc.pdf)]
As a result of this method, not all values are reported with the same number of decimal places. Given that this criterion is well-established and valid, and consistent with best practices in scientific reporting, the authors regret not making the change suggested by the reviewer.
Comment 3. The discussions are not convictive or strict, for example, line 361-362, TPC can react with reducing sugars? And why sugar was degraded during microwave heating?
Response 3. Thank you very much for your recommendations. Sorry for the misunderstanding, we did not mean that TPC reacts with sugars, but it is the Folin-Ciocalteau reagent which can react with sugars giving rise to a TPC value higher than the real one. Indeed, for sugar-rich products (mainly reducing sugars ones) a correction is recommended. To amend or clarify this point, the sentence has been replaced by “Additionally, since the Folin-Ciocalteau reagent can react with reducing sugars [32]…”. Reference [32] is the one to Singleton et al. (1999) [32], who proposed the Folin-Ciocalteau method for TPC determination, where these interferences are reported.
As for sugar degradation, hydrothermal degradation of sugars during MW treatment was explained in previous paragraphs in the original version of the manuscript (before Table 1), where it can be read: “Hydrothermal degradation of sugars is an extensively documented phenomenon and can occur during microwave treatment, as evidenced in studies on fruit residues [9]. This effect becomes more relevant when increasing power or treatment duration, since the simple sugars released due to local explosions or other phenomena can be subsequently degraded, leading to the generation of compounds which are inhibitory to microorganisms such as furfural derivatives […]”.
Comment 4. The conclusion section is too long, and the concise and accurate conclusion should be provided. Please reorganize this section.
Response 4. We appreciate Reviewer 4’s suggestion. In response, the Conclusion section has been reorganized and significantly shortened to present a more concise and focused summary of the main findings. It also now includes a brief outlook on future research directions. The revised section appears as follows:
“Overall, this study demonstrates that pretreatments can significantly influence the success of fermentation in terms of both microbial development and the retention or enhancement of bioactive compounds. Microwaving at 4 W/g for 5 min following pasteurization appears to be the most favorable option to obtain a fermented product with the highest phenolic content and microbial viability. In contrast, autoclaving is more effective when the goal is to maximize flavonoid concentration and antioxidant activity. These findings contribute valuable insight into the selection of appropriate pretreatments for maximizing the functional and nutritional qualities of fermented vegetable by-products.
Although 96-h fermentation with Lactiplantibacillus plantarum was not proved to consistently enhance the antioxidant properties of ground broccoli stalks, it is expected that further research focused on determining specific bioactive compounds or other fermentation products (phenolics, volatile compounds) will reveal valuable information about the success of fermentation after applying thermophysical pretreatments. It would also be interesting to explore the changes in the bioavailability of specific bioactive compounds as a result of the treatments applied to the plant matrix.
Maximizing the functional and nutritional qualities of fermented vegetable by-products can effectively contribute to food systems circularity by the valorization of discards and vegetable wastes. Therefore, the findings of the present research provide valuable insights towards the development of Sustainable Food Systems.”

Round 2
Reviewer 1 Report
Comments and Suggestions for Authors
Accepted
Reviewer 2 Report
Comments and Suggestions for Authors
The authors responded to my suggestions and they have addressed all the comments appropriately. In my opinion, the manuscript is now ready for acceptance.